# Feasibility and Acceptability of Integrating Acupuncture for Management of Multiple Symptoms in Medically Underserved Breast Cancer Survivors

**DOI:** 10.3390/cancers17020304

**Published:** 2025-01-18

**Authors:** Hongjin Li, Ardith Z. Doorenbos, Zhengjia Chen, Hannah Choi, Weiwei Ma, Oana Danciu, Crystal L. Patil, Shuang Gao, Natalie Lif, Judith M. Schlaeger

**Affiliations:** 1College of Nursing, University of Illinois Chicago, Chicago, IL 60612, USA; ardith@uic.edu (A.Z.D.); hchoi213@uic.edu (H.C.); nlif2@uic.edu (N.L.); jschlaeg@uic.edu (J.M.S.); 2University of Illinois Cancer Center, Chicago, IL 60612, USA; znchen@uic.edu (Z.C.); wma29@uic.edu (W.M.); ocdanciu@uic.edu (O.D.); 3School of Nursing, University of Michigan, Ann Arbor, MI 48109, USA; clpatil@med.umich.edu; 4College of Medicine, University of Illinois Chicago, Chicago, IL 60612, USA; sgao20@uic.edu

**Keywords:** acupuncture, health disparities, medically underserved, breast cancer survivors, symptom management

## Abstract

Breast cancer survivors often experience multiple symptoms such as pain, fatigue, sleep disturbance, depression, anxiety, and hot flashes during endocrine therapy. This study examined the feasibility of integrating acupuncture into treatment options for symptom management among medically underserved breast cancer survivors. We randomized 62 participants into acupuncture and usual care groups. High retention and satisfaction rates indicated that acupuncture is both feasible and acceptable for this population. After 10 acupuncture sessions over 5 weeks, participants showed significant improvements in pain, fatigue, sleep disturbance, depression, and anxiety. This approach shows potential for reducing symptoms and addressing health disparities among underserved breast cancer survivors.

## 1. Introduction

Breast cancer remains the most common cancer among women worldwide. An estimated 310,000 new cases of invasive breast cancer are expected to be diagnosed in the United States (U.S.) in 2024, contributing to the growing population of 4.1 million breast cancer survivors [1]. With advances in early detection and treatment, more women are surviving breast cancer. Long-term survivorship, however, often comes with a significant symptom burden [2,3]. Breast cancer survivors often experience a range of distressing symptoms, particularly during long-term endocrine therapy, which is commonly prescribed to reduce the risk of cancer recurrence [4,5]. A recent systematic review revealed that the most common symptoms breast cancer survivors report were joint pain (64%), hot flashes (75%), fatigue (80%), and sleep disturbance (50%) [6]. These symptoms can persist long after treatment ends [7], profoundly impacting breast cancer survivors’ quality of life (QOL) and ability to work [8,9]. Therefore, effective, individualized symptom management is crucial, especially during long-term endocrine therapy, as adherence to this therapy is essential for survival [4].

Acupuncture, a modality of Traditional Chinese Medicine, has emerged as a promising nonpharmacological intervention for managing multiple cancer-related symptoms [10]. Research has demonstrated its potential to reduce pain, improve sleep, reduce fatigue, and improve other cancer-related symptoms [11]. Medically underserved breast cancer survivors tend to have greater pain and greater mental health distress [12,13]. However, a review indicated that this population experiences disparities in the receipt of survivorship care plans and has substantial unmet needs with respect to their survivorship care [14]. They often face barriers to accessing comprehensive care and supportive therapies [15,16], including acupuncture. Financial constraints, cultural differences, lack of insurance coverage, and logistical challenges, such as transportation and clinic availability, often prevent medically underserved cancer survivors from benefiting fully from complementary and integrative health (CIH) therapies [17]. Expanding access to acupuncture for underserved populations is crucial to providing equitable care and addressing disparities in symptom management for breast cancer survivors [17].

In the United States, most acupuncture trials have been conducted in primarily non-Hispanic White populations, and acupuncture services are often concentrated in clinics that cater to more affluent or insured individuals [17]. To our knowledge, only a few studies have explored the integration of acupuncture into the care of patients from underserved communities, focusing mainly on chronic pain [18]. There is little research on its use for medically underserved breast cancer survivors. Addressing these gaps is critical, exploring the feasibility and acceptability of acupuncture in this population may offer a promising avenue for alleviating symptoms and improving well-being, while also informing strategies to enhance access and equity in survivorship care. In the present pilot study, the primary aim is to assess the feasibility and acceptability of integrating acupuncture for the treatment of multiple symptoms into clinics serving medically underserved breast cancer survivors. The secondary aim is to assess changes in symptom outcomes and compare these changes between the acupuncture group and the usual care group at Week 6 and Week 12.

## 2. Materials and Methods

### 2.1. Participants and Study Design

In this randomized controlled trial (RCT), early-stage breast cancer survivors were assigned to receive either a 5-week acupuncture treatment or usual care (control). Participants were recruited from the Mile Square Health Center (MSHC) Survivorship Clinic and the University of Illinois Hospital and Health Sciences (UI Health) Outpatient Care Center Oncology Clinic between January 2023 and August 2024. Both the UI Oncology Clinic and MSHC serve an underserved and diverse population. The MSHC is a federally qualified health center (FQHC), with most patients being racial and ethnic minority women and children who live at or below the poverty line and are publicly insured. This study was registered with Clinical Trials.gov (NCT05615753). Human subject approval was obtained from the University of Illinois Chicago’s Human Research Protection Office (IRB no. 2022-0353) [19]. This study was designed in accordance with the CONSORT 2010 Statement (Appendix A) [20].

Eligible participants were women who (a) had histologically confirmed Stage 0, I, II, or III breast cancer; (b) were at least 18 years old; (c) had completed their primary cancer treatment (i.e., surgery, radiotherapy, chemotherapy) within 1 month prior to the start of the study and were currently being treated with endocrine therapy; (d) were able to speak and read English; and (e) self-reported experiencing pain, fatigue, sleep disturbance, depression, anxiety, and/or hot flashes in the last month. Specifically, eligible women had to report a minimum average severity rating of 3 or greater (0–10 numeric rating scale) for at least three of the six symptoms, and at least two of the symptoms had to be physical (pain, fatigue, or sleep disturbance). Participants were excluded if they (a) had metastatic breast cancer (Stage IV); (b) had a bleeding disorder (e.g., hemophilia, Von Willebrand disease, thrombocytopenia); (c) did not have time to attend the twice weekly acupuncture sessions for 5 consecutive weeks within the next few months; (d) had psychiatric or medical disorders that would affect study assessments such as dementia, Alzheimer’s disease, traumatic brain injury, stroke, or history of any neurological condition; or (e) were breastfeeding or pregnant or were planning on becoming pregnant during the study period.

### 2.2. Procedure

#### 2.2.1. Recruitment

The research coordinator (RC) reviewed patients’ electronic health records (EHRs) to identify survivors who met the inclusion criteria. The RC then reached out to eligible breast cancer survivors via email and text message providing study information along with an attached flyer. Additionally, the RC informed providers when eligible survivors were scheduled for routine visits at the clinic. During these visits, providers introduced the eligible breast cancer survivors to the study using a recruitment script and the study flyer. Providers asked about survivors’ interest in participating or for permission for the RC to contact them.

#### 2.2.2. Screening

The RC then contacted eligible survivors to provide further details about the study and conduct a more in-depth assessment to confirm their eligibility for participation The RC conducted screening interviews in the clinic waiting room or by phone with survivors who expressed an interest in participating, asking them to rate their level of pain, fatigue, sleep disturbance, depression, anxiety, and hot flashes over the past 1 month on a scale of 0–10. Those who met the eligibility criteria and agreed to participate were then asked to provide written informed consent and scheduled for the baseline (Week 0) study visit.

#### 2.2.3. Data Collection

Participants completed all outcome assessments in REDCap [21], thus enabling a form of blinded outcomes assessment. At the baseline study visit (Week 0), the RC instructed participants on completing study questionnaires in REDCap [21] and asked them to complete the demographic and patient-reported outcomes questionnaires. Participants received email reminders to complete the study questionnaires for Weeks 6 and 12 through REDCap 2 days before the due dates. The RA checked for completed questionnaires on the due date and as needed, sent a second email reminder with a follow-up phone call to ensure that the questionnaires were completed within 3 days.

#### 2.2.4. Randomization

After all the participants completed the baseline questionnaires in REDCap, they were randomly assigned 1:1, via REDCap’s randomization module. The allocation was stratified by age and baseline composite symptom score and followed a permuted block design.

### 2.3. Intervention

The acupuncture intervention involved 10 sessions delivered twice per week over a 5-week period, with at least one day separating each session. A semi-standardized protocol was used for all participants, which included (1) standardized acupuncture points aimed at managing symptoms such as pain, fatigue, sleep disturbances, depression, anxiety, and hot flashes; and (2) additional points customized to address up to three of the participant’s most painful areas (e.g., chest, fingers, shoulders, lower back, knees, hips, wrists, feet, or toes), which could be adjusted at each session. Needles were kept in place for 30 min and gently rotated before and after insertion. The acupuncture needles used were all 0.25 mm diameter x 40 mm length. Details of the acupuncture points are available in the published protocol [19]. The acupuncture intervention was conducted by three acupuncturists at the Acupuncture Research Lab of the University of Illinois Chicago College of Nursing near MSHC and UI Health.

### 2.4. Usual Care Group

Participants in the usual care group were given written educational resources on managing breast cancer symptoms and maintained their standard treatments (endocrine therapy) as prescribed. No additional interventions or acupuncture treatments were offered to these participants during the course of the study.

### 2.5. Measures

#### 2.5.1. Demographic and Clinical Characteristics

Sociodemographic data (age, race, education, occupation, family income, household size, marital status) were collected at baseline through a questionnaire. Clinical information, such as disease stage and treatment types, was obtained from the EHR.

#### 2.5.2. Symptoms

The PROMIS-29 (v1.0) is a 29-item, generic health assessment tool that measures multiple domains, including physical functioning, anxiety, depression, fatigue, sleep disturbance, and pain interference. Each domain comprises four items, and each item is rated on a 5-point Likert scale (from 1 to 5). It also contains an extra item to assess pain intensity, which respondents rank on a scale of 1 (“no pain”) to 10 (“worst imaginable pain”). The PROMIS-29 is a reliable measure of symptoms among cancer patients with good internal consistency and convergent validity (Cronbach’s alphas: 0.81–0.96) [22,23,24]. The PROMIS scores are reported on the T-score metric, which allows for comparison to population norms. In this scoring system, a T-score of 50 represents the mean for the general population, and each 10-point increment corresponds to one standard deviation. Higher scores reflect a greater degree of the construct or severity of the symptom being measured.

The hot flash composite score (HFCS) was calculated by multiplying the average daily frequency of hot flashes in the preceding week by the average severity rating [25]. Participants documented the frequency and intensity of hot flashes at baseline, Week 6, and Week 12 using the Daily Hot Flash Diary, rating severity on a scale from 1 (“mild”) to 4 (“very severe”). The HFCS has been established as a reliable and valid measure in previous acupuncture studies involving women with breast cancer [26].

Since the primary symptom outcome for this study is the symptom cluster, we calculated a symptom cluster composite score by averaging participants’ scores for pain, fatigue, sleep disturbance, depression, anxiety, and hot flashes.

#### 2.5.3. Feasibility and Acceptability

Feasibility outcomes include the following:(a)Recruitment rate: calculated as the number of randomized women out of those we screened and contacted.(b)Retention rate: calculated as the number of women who completed baseline and follow-up assessments at week 6 and week 12 out of the total number of participants.(c)Engagement/adherence rate: calculated as the number of women who completed all 10 acupuncture sessions out of those randomized to the acupuncture group.(d)Fidelity: calculated as the number of times acupuncturists followed the treatment protocol and adhered to the standardized point selection guidelines.

The Protocol Acceptability Scale is a 9-item self-report measure that assesses study acceptability. It has exhibited stable test-retest reliability in prior studies [27,28]. Items are measured on a scale from 0 (negative response) to 2 (positive response), with total scores ranging from 0 to 18. Higher scores indicate greater levels of acceptability. We administered the scale to participants in the acupuncture group in Week 6 after acupuncture treatment. The study protocol was deemed to have high acceptability if 80% of participants reported a total score of ≥14.5.

To ensure the integrity and consistency of the acupuncture intervention, fidelity checks were conducted throughout the study. Fidelity was assessed across several key dimensions, including (1) adherence to the treatment protocol, (2) standardization of acupuncture point selection, and (3) consistency in session duration and frequency. Random unscheduled fidelity checks were performed approximately every 3 months. One of our investigators (J.S.) monitored the needle insertions for appropriate placement during training and fidelity checks.

### 2.6. Data Analysis

Demographic and clinical characteristics and symptom measures were summarized as mean (standard deviation) or count (percentage) for continuous and categorical variables, respectively. To make the HFCS comparable to the PROMIS measures, the HFCS was standardized to have a distribution of mean at 50 and standard deviation at 10, with a range from 0 to 100 (measurements that were −5 SD from the mean were set to be 0, and those 5 SD from the mean were set to be 100). Baseline comparisons between the acupuncture and control groups were conducted using the Wilcoxon rank sum test (Mann–Whitney U test) for numeric variables and Fisher’s exact test for categorical variables. Because our outcome measures were repeated measures over time, we assessed differences in changes from baseline to Week 6 and Week 12 using mixed-effect models [29]. Fixed-effect factors associated with the outcomes of interest included group assignment (acupuncture versus usual care group), time (treated as a categorical variable), and intervention × time interaction. Results are presented as between-group differences from baseline to Week 6 and Week 12 with 95% confidence intervals. The analyses were conducted using SAS 9.4 (SAS Institute, Cary, NC) [30]. All tests were two-sided, and *p*-values < 0.05 were considered significant.

For the feasibility testing, we aimed to detect recruitment, retention, completion, and acceptability rates using a 95% two-sided confidence interval. Assuming retention rates are within the desired range of 80% or higher, we require at least 62 breast cancer survivors to achieve a 95% confidence level (from 67.8% to 89.1%).

## 3. Results

### 3.1. Demographic and Clinical Characteristics

Sixty-two patients were randomly assigned to the acupuncture intervention (*n* = 31) and usual care (*n* = 31) groups (Figure 1). Table 1 summarizes the participants’ demographic and clinical characteristics at Week 0. Participants had an average age of 55.2 ± 9.3 years and a BMI of 31.6 kg/m^2^; the majority had attained at least a bachelor’s degree (51.5%) and had a household income of less than $55,000 (62.9%). The study population was diverse, with the majority (54.8%) identifying as Black or African American. All participants were diagnosed with early-stage breast cancer, with nearly half (46.8%) classified as Stage 1. All participants were receiving endocrine therapy; drug type varied, with the greatest number (35.5%) taking anastrozole. The top three pain sites were the hand joints (62.9%), the knee joints (61.3%), and the lower back (51.6%). Nearly one-third of participants (32.3%) reported antidepressant use. The most common comorbidity was high blood pressure (50.0%). There were no differences between groups on demographic or clinical characteristics at baseline.

### 3.2. Feasibility

#### 3.2.1. Recruitment and Retention

We contacted and screened 300 potential participants. Among them, 62 women were interested in participating and provided informed consent. After consent and randomization, two participants dropped out because of lack of time and illness of family members.

A total of 58 participants (93.5%) completed the Week 6 assessment, and 56 (90.3%) completed the Week 12 assessment (Figure 1). The study attrition rate was 9.7%, which was less than our target of 20%.

#### 3.2.2. Engagement

Most participants in the acupuncture group (93.1%) completed all 10 sessions of their acupuncture treatment plan. For the four participants who missed acupuncture sessions, the primary reasons included pain, shoulder problems, illness or hospitalization, and scheduling conflicts with other appointments. These barriers highlight the health and logistical challenges breast cancer survivors face.

#### 3.2.3. Fidelity

All three acupuncturists followed the treatment protocol and adhered to the standardized point selection guidelines. Deviations in session frequency or duration were documented after each session to maintain accuracy in data collection and intervention delivery.

### 3.3. Acceptability and Survivor Treatment Satisfaction

Table 2 shows the feasibility and acceptability responses. Among those who completed the Protocol Acceptability Scale for Treating Psychoneurological Symptoms with Acupuncture (*n* = 27), the mean acceptability score was 16.6 ± 1.4, with 92.8% of respondents rating it as acceptable (score > 14.5). This result met our criterion of an acceptability rate of 80% or more.

The satisfaction assessment survey was completed by 27 of the 31 participants who received acupuncture. All expressed that they liked acupuncture, with 63% being very satisfied with their treatment and 29.6% satisfied. Additionally, 66.7% strongly agreed that the acupuncture intervention was beneficial and worth their time. Half of the participants (50%) strongly agreed, and 23.1% agreed that acupuncture helped them manage their cancer-related symptoms.

### 3.4. Symptom Outcomes

There were no significant differences in the severity of pain interference, fatigue, sleep disturbance, depression, anxiety, or hot flashes between the two groups at baseline (Table 3). As shown in Table 4, compared to participants receiving usual care, participants who received acupuncture treatment had significantly greater reductions in pain, fatigue, sleep disturbance, and depression as well as in the composite score of six symptoms at both Week 6 and Week 12. Anxiety was significantly reduced in the acupuncture group compared to the control group at Week 6 only. There were no significant differences in the reduction in hot flashes between the two groups at either time point.

## 4. Discussion

This study evaluated the feasibility and acceptability of a 5-week acupuncture intervention for the treatment of multiple cancer-related symptoms in medically underserved breast cancer survivors during endocrine therapy. This study is important because most prior acupuncture trials have focused on predominantly White populations, leaving a gap in knowledge of the feasibility and acceptability of this intervention among diverse and underserved groups. Participants in the acupuncture group showed significant reductions in the symptom cluster and multiple symptoms, including pain interference, fatigue, sleep disturbances, depression, and anxiety compared to those receiving usual care.

We found that the integration of acupuncture into clinics caring for medically underserved breast cancer survivors is both feasible and acceptable. The high recruitment, retention, engagement, and satisfaction rates for this study suggest there is demand for acupuncture treatment among this population. Many medically underserved breast cancer survivors voluntarily asked to participate in the study to receive acupuncture treatment. Successful recruitment is highly attributed to the referrals from oncologists, who play a pivotal role in bridging the gap between patients and CIH therapies. This emphasizes the importance of integrating oncologist referrals into recruitment strategies for future implementation studies, particularly those focusing on underserved populations. Additionally, the willingness of participants to join the study and their adherence to the treatment underscores the unmet need for acupuncture to address multiple treatment-related symptoms often experienced by breast cancer survivors. When barriers such as cost and availability are mitigated, patients are likely to engage with and benefit from acupuncture, highlighting the importance of improving access to acupuncture in medically underserved breast cancer survivors.

Several challenges encountered during the implementation of this project may offer valuable insights for future research efforts. Black women are more likely than non-Black women to be diagnosed with triple-negative breast cancer, which does not require endocrine therapy. During screening, we noticed that many breast cancer survivors who were never prescribed endocrine therapy also reported experiencing multiple symptoms (e.g., pain, fatigue, hot flashes). Future studies should broaden the inclusion criteria to extend beyond breast cancer survivors receiving endocrine therapy. Expanding the scope to include a more diverse population of breast cancer survivors will provide a more comprehensive understanding of symptom management needs across different subgroups, allowing for more inclusive and tailored interventions.

A barrier to providing acupuncture in medically underserved populations is the lack of insurance reimbursement. Our study offered free acupuncture and successfully demonstrated the feasibility of delivering care through this model. Many participants expressed a strong desire to continue receiving acupuncture after completing the study, but the out-of-pocket cost and lack of insurance reimbursement were significant barriers. To address this challenge, we have explored and will continue to pursue, various strategies to make acupuncture more affordable and accessible. In April 2023, Illinois Medicaid began covering acupuncture for chronic low back pain and breech presentation [31]. Findings from the present study provide evidence that can be used to support expanding acupuncture coverage for medically underserved breast cancer survivors. More equitable access to healthcare is needed to improve treatment options for this population.

In addition, participants in the acupuncture group showed significant reductions in multiple symptoms, including pain interference, fatigue, sleep disturbances, depression, and anxiety compared to those receiving usual care. Notably, most of these symptom improvements persisted for up to 6 weeks post-intervention, except for anxiety. Published estimates for the Minimally Important Change (MIC) for PROMIS measures do vary widely depending on the specific domain and population studied. A systematic review of MIC estimates for PROMIS measures suggests a range of 2 to 6 T-score points for a MIC [32]. In our study, most of the symptoms exhibited a clinically significant change after acupuncture based on the MIC estimates for PROMIS. In this study, acupuncture appeared to have no significant effects on hot flashes. Findings are partially consistent with previous meta-analyses [11,33,34] reporting that acupuncture significantly reduced pain, fatigue, sleep disturbances, and hot flashes compared to usual care in patients with breast cancer. One possible explanation for the inconsistent results regarding anxiety and hot flashes is our inclusion criteria: to be eligible for participation, women had to present with at least three out of six symptoms, and at least two had to be physical symptoms (such as pain, fatigue, or sleep disturbance). Consequently, some participants may not have experienced moderate or severe hot flashes or anxiety at baseline, which could have affected our ability to detect the effects of acupuncture on these two symptoms.

Our study has some limitations. First, we recruited breast cancer survivors from two clinics that primarily serve medically underserved populations in Chicago. Randomization was not stratified by clinics to control potential confounding factors. Future studies should consider stratified randomization by clinics to better control for potential confounders. Second, it is important to note that we did not include Spanish-speaking breast cancer survivors. Findings may thus not be fully representative of the broader population of medically underserved breast cancer survivors, particularly those from different linguistic, cultural, or ethnic groups. Results, therefore, may not be generalizable to other health systems or patient populations with different demographic characteristics or language needs. Future research should include multiple sites across various settings and patient populations to improve the generalizability of the findings. Third, participants with an interest in acupuncture were more likely to join and adhere, resulting in non-random attrition. This introduces selection bias and limits the generalizability of findings to the broader population of medically underserved breast cancer survivors. Future studies should consider conducting drop-out analyses to assess whether attrition is non-random. Another limitation is the usual care group did not receive special attention. Future studies should include an attention-controlled group to have 10 equally long visits without acupuncture to control for the potential effects of attention and psychosocial support. Also, long-term follow-ups in clinical studies can be challenging, and in this study, the 12-week follow-up did not completely account for potential changes in medical treatment, such as increased analgesic use, which could have influenced symptom assessments, particularly for pain. Lastly, the intervention was delivered in a nearby integrative clinic. To improve convenience and accessibility for patients, future studies should explore the feasibility of delivering acupuncture treatment within the oncology clinic or an FQHC and integrating acupuncture services into the care plan of underserved breast cancer survivors. This shift could enhance the integration of acupuncture into conventional care settings, allowing for better coordination among healthcare providers and more streamlined access for patients.

## 5. Conclusions

Our findings suggest that integrating acupuncture into care to manage multiple cancer-related symptoms is feasible and acceptable in outpatient clinics serving medically underserved breast cancer survivors. To further validate these results and assess their applicability across different settings and survivor populations, including non-English-speaking populations, a multi-site implementation trial is needed. Such a trial would provide a more comprehensive understanding of acupuncture’s efficacy in managing common symptoms reported by breast cancer survivors across diverse health systems and patient populations in the U.S.

## Figures and Tables

**Figure 1 cancers-17-00304-f001:**
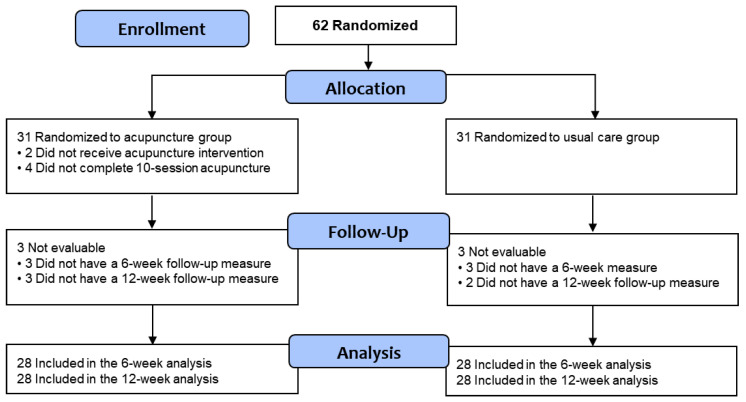
CONSORT flow diagram.

**Table 1 cancers-17-00304-t001:** Baseline demographic and clinical characteristics.

Characteristic	Total (*N* = 62)	Acupuncture (*n* = 31)	Usual Care (*n* = 31)	*p*-Value
**Age (years)**	55.2 (9.3)	55.4 (9.1)	55.1 (9.5)	0.921
**Ethnicity**				0.534
Hispanic/Latino	13 (21.0%)	8 (25.8%)	5 (16.1%)	
Non-Hispanic	48 (77.4%)	23 (74.2%)	25 (80.7%)	
Unknown	1 (1.6%)	0 (0.0%)	1 (3.2%)	
**Race**				0.620
Black/African American	34 (54.9%)	15 (48.4%)	19 (61.3%)	
White/Caucasian	18 (29.0%)	11 (35.5%)	7 (22.5%)	
Asian	3 (4.8%)	1 (3.2%)	2 (6.5%)	
American Indian/Alaska Native	1 (1.6%)	1 (3.2%)	0 (0.0%)	
Unknown/other	6 (9.7%)	3 (9.7%)	3 (9.7%)	
**Education level**				0.942
<High school	4 (6.5%)	2 (6.5%)	2 (6.5%)	
High school graduate	9 (14.5%)	4 (12.9%)	5 (16.1%)	
Some college	17 (27.4%)	7 (22.5%)	10 (32.3%)	
Bachelor’s degree	19 (30.6%)	11 (35.5%)	8 (25.8%)	
Some graduate school	2 (3.2%)	1 (3.2%)	1 (3.2%)	
Graduate degree	11 (17.7%)	6 (19.4%)	5 (16.1%)	
**Marital status**				0.723
Never married	19 (30.6%)	8 (25.8%)	11 (35.5%)	
Married/living w/partner	21 (33.9%)	10 (32.3%)	11 (35.5%)	
Divorced/separated	17 (27.4%)	10 (32.3%)	7 (22.5%)	
Widowed	5 (8.1%)	3 (9.7%)	2 (6.5%)	
**Annual household income**				0.936
<$35,000	23 (37.1%)	12 (38.7%)	11 (35.5%)	
$35,000–$54,999	16 (25.8%)	8 (25.8%)	8 (25.8%)	
$55,000–$100,000	15 (24.2%)	8 (25.8%)	7 (22.5%)	
>$100,000	8 (12.9%)	3 (9.7%)	5 (16.1%)	
**Cancer stage**				0.674
0	2 (3.2%)	1 (3.2%)	1 (3.2%)	
I	29 (46.8%)	14 (45.2%)	15 (48.4%)	
II	22 (35.5%)	13 (41.9%)	9 (29.0%)	
III	9 (14.5%)	3 (9.7%)	6 (19.4%)	
**Treatment history ^a^**				
Chemotherapy	40 (64.5%)	20 (64.5%)	20 (64.5%)	1.000
Radiation therapy	50 (82.0%)	24 (77.4%)	26 (83.9%)	0.749
**Endocrine therapy drug type**				0.321
Anastrozole	22 (35.5%)	14 (45.2%)	8 (25.8%)	
Exemestane	9 (14.5%)	5 (16.1%)	4 (12.9%)	
Letrozole	19 (30.6%)	8 (25.8%)	11 (35.5%)	
Tamoxifen	12 (19.4%)	4 (12.9%)	8 (25.8%)	
**Pain site ^a^**				
Joints, hands	39 (62.9%)	21 (67.7%)	18 (58.1%)	0.600
Joints, knees	38 (61.3%)	20 (64.5%)	18 (58.1%)	0.795
Joints, feet/ankles	27 (43.5%)	13 (41.9%)	14 (45.2%)	1.000
Lower back	32 (51.6%)	16 (51.6%)	16 (51.6%)	1.000
Legs	24 (38.7%)	15 (48.4%)	9 (29.0%)	0.192
Arms	28 (45.2%)	15 (48.4%)	13 (41.9%)	0.799
Breast/chest	30 (48.4%)	16 (51.6%)	14 (45.2%)	0.800
**Antidepressant use, yes**	20 (32.3%)	12 (38.7%)	8 (25.8%)	0.416
**Comorbidity ^a^**				
Heart disease	4 (6.5%)	2 (6.5%)	2 (6.5%)	1.000
Diabetes	12 (19.4%)	6 (19.4%)	6 (19.4%)	1.000
High blood pressure	31 (50.0%)	15 (48.4%)	16 (51.6%)	1.000
Liver disease	1 (1.6%)	0 (0.0%)	1 (3.2%)	1.000
Kidney disease	1 (1.6%)	1 (3.2%)	0 (0.0%)	1.000
**BMI (kg/m^2^)**	32.0 (6.4)	32.4 (6.2)	31.6 (6.7)	0.647

^a^ Respondents were permitted to select multiple answers. Data are provided as *n* (%) for categorical variables and mean (standard deviation) for continuous. BMI, body mass index.

**Table 2 cancers-17-00304-t002:** Feasibility and acceptability responses.

Domain	Items	Mean (SD)
Feasibility	Recruitment rate	20.7%
	Retention rate	90.3%
	Engagement/adherence rate	93.1%
	Fidelity	100%
Acceptability	1. Was participating in this study too hard?	1.9 (0.4)
	2. Were the study instructions easy to understand?	1.9 (0.3)
	3. Did you feel rushed to complete this study?	2.0 (0.0)
	4. Did you enjoy being in this study?	1.9 (0.3)
	5. What do you think of getting acupuncture?	1.8 (0.6)
	6. Did you think acupuncture was painful?	1.8 (0.4)
	7. Would you get acupuncture again?	2.0 (0.0)
	8. Do you think this study will be well received by other breast cancer survivors?	1.7 (0.4)
	9. Do you think the length of the study period was	1.6 (0.6)
	Total acceptability score	16.6 (1.4)
	Acceptability rate	92.8%

SD, standard deviation.

**Table 3 cancers-17-00304-t003:** Baseline symptom severity (Week 0): comparison between the two groups.

Symptom	Total (*N* = 62)	Acupuncture (*n* = 31)	Usual Care (*n* = 31)	*p*-Value ^a^
Pain interference ^b^	58.9 (8.5)	59.6 (8.5)	58.2 (8.5)	0.5014
Fatigue ^b^	57.1 (8.2)	57.7 (8.0)	56.5 (8.5)	0.5766
Anxiety ^b^	55.7 (8.3)	55.6 (8.1)	55.7 (8.6)	0.9410
Depression ^b^	52.2 (8.6)	53.7 (8.4)	50.8 (8.6)	0.1811
Sleep disturbance ^b^	57.9 (8.6)	58.4 (6.1)	57.4 (10.6)	0.6681
Hot flash composite score	50.0 (6.9)	49.1 (2.2)	50.9 (9.5)	0.3199
Symptom cluster composite score ^c^	55.3 (5.3)	55.7 (4.3)	54.9 (6.2)	0.5781

^a^ Two-sample *t*-test; ^b^ measured with the PROMIS-29 (v1.0) questionnaire; ^c^ the symptom cluster composite score was calculated by average the scores for pain, hot flashes, fatigue, sleep disturbance, depression, and anxiety.

**Table 4 cancers-17-00304-t004:** Changes in symptom severity from baseline to Week 6 and Week 12: Comparison between the two groups.

	Mean Change from Baseline (95% CI)		Between-Group Difference (95% CI)	
	Acupuncture (*N* = 28)	Usual Care (*N* = 28)	Acupuncture vs. Usual Care	*p*-Value ^a^
Pain interference ^b^				
Week 6	−5.1 (−7.3 to −2.8)	−0.1 (−2.2 to 2.0)	−4.9 (−8.0 to −1.9)	**0.0018**
Week 12	−6.3 (−8.5 to −4.1)	−0.9 (−3.1 to 1.3)	−5.4 (−8.5 to −2.3)	**0.0008**
Fatigue ^b^				
Week 6	−9.3 (−11.8 to −6.8)	−0.8 (−3.2 to 1.6)	−8.5 (−12.0 to −5.0)	**<0.0001**
Week 12	−5.7 (−8.2 to −3.2)	0.9 (−1.6 to 3.3)	−6.6 (−10.1 to −3.1)	**0.0003**
Anxiety ^b^				
Week 6	−4.8 (−7.4 to −2.2)	2.5 (0.02 to 4.9)	−7.3 (−10.8 to −3.7)	**<0.0001**
Week 12	−1.5 (−4.0 to 1.0)	−0.3 (−2.8 to 2.2)	−1.2 (−4.8 to 2.3)	0.4988
Depression ^b^				
Week 6	−5.2 (−7.7 to −2.7)	−0.6 (−3.0 to 1.8)	−4.6 (−8.1 to −1.1)	**0.01**
Week 12	−2.7 (−5.2 to −0.2)	2.5 (−0.004 to 4.9)	−5.1 (−8.6 to −1.6)	**0.0043**
Sleep disturbance ^b^				
Week 6	−7.1 (−10.1 to −4.2)	−0.9 (−3.7 to 1.9)	−6.2 (−10.2 to −2.2)	**0.003**
Week 12	−5.5 (−8.4 to −2.7)	−0.8 (−3.6 to 2.1)	−4.8 (−8.8 to −0.7)	**0.022**
Hot flash composite score				
Week 6	−1.1 (−2.4 to 0.2)	−0.5 (−1.8 to 0.7)	−0.6 (−2.4 to 1.2)	0.5096
Week 12	−1.0 (−2.3 to 0.2)	−0.1 (−1.4 to 1.2)	−0.9 (−2.7 to 0.9)	0.3138
Symptom cluster composite score ^c^				
Week 6	−5.4 (−6.8 to −4.1)	−0.1 (−1.5 to 1.2)	−5.3 (−7.3 to −3.4)	**<0.0001**
Week 12	−3.8 (−5.2 to −2.4)	0.2 (−1.2 to 1.5)	−4.0 (−5.9 to −2.0)	**0.0001**
Pain/fatigue/sleep disturbance symptom cluster ^d^				
Week 6	−7.2 (−9.1 to −5.3)	−0.6 (−2.4 to 1.2)	−6.6 (−9.2 to −4.0)	**<0.0001**
Week 12	−5.9 (−7.7 to −4.0)	−0.3 (−2.1 to 1.5)	−5.6 (−8.2 to −3.0)	**<0.0001**

^a^ *p*-values were calculated using the mixed-effects model; bolded are *p*-value < 0.05. Data are presented as mean (95% confidence interval). ^b^ Measured with the PROMIS-29 (v1.0) questionnaire; ^c^ the symptom cluster composite score was calculated by averaging the scores for pain interference, hot flashes, fatigue, sleep disturbance, depression, and anxiety. ^d^ The symptom cluster composite score was calculated by averaging the scores for pain interference, fatigue, and sleep disturbance.

## Data Availability

Full trial protocol can be accessed at https://doi.org/10.1016/j.cct.2023.107387 (accessed on 17 December 2024). Participants’ survey data are available upon request to interested researchers.

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
