# Peer review of "Feasibility and Acceptability of Integrating Acupuncture for Management of Multiple Symptoms in Medically Underserved Breast Cancer Survivors"

_cancers, 2025, doi:10.3390/cancers17020304_

Round 1

Reviewer 1 Report (Previous Reviewer 1)

Comments and Suggestions for Authors

Thanks to the authors for taking the time in considering all reviewers’ comments and thus, significantly improve the well-deserved quality of this research. It is also worth noting that authors dug deeper into the statistical analyses performed, as the results presented are now very well-articulated.

Reviewer 2 Report (Previous Reviewer 2)

Comments and Suggestions for Authors

Dear authors
all my previous remarks have been adequately addressed, Thus, I have no further comments.

This manuscript is a resubmission of an earlier submission. The following is a list of the peer review reports and author responses from that submission.

Round 1

Reviewer 1 Report

Comments and Suggestions for Authors

Thanks to the authors for undertaking this very interesting and health-oriented research on using acupuncture for reducing multiple symptoms in medically underserved breast cancer survivors. After a careful reading of your paper, I have found certain inconsistencies and questionable aspects of it that I would like authors to address thoughtfully.

1.      [Lines 26, 266, and 277] Although it is clear to me the measure (and use) of acceptability, I don’t understand how feasibility is measured, as conceptually means that an instrument can be applied, nothing else. Could the authors shed some light on this?

2.      [Lines 28, 86, and 87] The authors undertook the research based on a randomized sampled population taken from two different clinics. Did the authors verify that the controlled group had the same usual treatment in both clinics? How can the authors verify that the use of two different clinics is not a factor of interest?

3.      There is an inconsistency with the methods proposed for the statistical analyses and the ones presented in the results section (Tables 3 and 4). While authors claim to have used Mann-Whitney U test (line 216), the tables present results as two-sample T-tests.

4.      Following point 3 above, authors used Fisher’s exact test to deal with non-numerical variables (proportions). The issue here, is that in Table 2 a p-value is presented as overall comparison among categories for each group. Although in this is case is not relevant (as all p-values are greater than 0.05), I remind authors that this test assesses the existence (or not) of a difference among categories of the same variable but does not tell where that difference lays on.

5.      Provided point 3 above, one of my main concerns with this study is that authors undertook a longitudinal study. Thus, the appropriate tool to use to compare among groups and timeframes is a between-within analysis of variance (or the less powerful strategy Bonferroni Correction). Could the authors argue about this?

6.      My second main concern is that authors state that their findings suggest that “breast cancer survivors benefited from acupuncture…” [Line 303]. The statistical analysis performed assesses only differences, and not if one method is better than the other. I believe that authors missed to explicitly include their statistical hypotheses and the calculation of the corresponding effect sizes.

Author Response

  1. [Lines 26, 266, and 277] Although it is clear to me the measure (and use) of acceptability, I don’t understand how feasibility is measured, as conceptually means that an instrument can be applied, nothing else. Could the authors shed some light on this?

Response: Thank you for your comment. We agree that feasibility requires a clear explanation of how it is measured. In our study, feasibility is assessed through three key metrics: 1) recruitment; 2) retention, 3) engagement/adherence, and 4) fidelity (Pg 5, Lines 200-208).

  1. [Lines 28, 86, and 87] The authors undertook the research based on a randomized sampled population taken from two different clinics. Did the authors verify that the controlled group had the same usual treatment in both clinics? How can the authors verify that the use of two different clinics is not a factor of interest?

Response: In our study, the usual treatment was endocrine therapy, which was an inclusion criterion for all participants. To ensure consistency, we confirmed that all participants, regardless of the clinic, were receiving endocrine therapy as part of their standard care.

We acknowledge that the use of two different clinics could introduce variability. To address this, we have highlighted this as a limitation in our manuscript and recommend future studies adopt stratified randomization by clinic to better control for potential confounders (Pg 10, Lines 396-401).

  1. There is an inconsistency with the methods proposed for the statistical analyses and the ones presented in the results section (Tables 3 and 4). While authors claim to have used Mann-Whitney U test (line 216), the tables present results as two-sample T-tests.

Response: Thank you for your careful review and for pointing out the inconsistency. We have revised the Data Analysis section in the Methods to clarify that the Mann-Whitney U test was used to compare the acupuncture and control groups on demographic and clinical characteristics at baseline, as presented in Table 2. These revisions ensure alignment between the described methods and the results (Pg 5, Lines 229-233).

  1. Following point 3 above, authors used Fisher’s exact test to deal with non-numerical variables (proportions). The issue here, is that in Table 2 a p-value is presented as overall comparison among categories for each group. Although in this is case is not relevant (as all p-values are greater than 0.05), I remind authors that this test assesses the existence (or not) of a difference among categories of the same variable but does not tell where that difference lays on.

Response: Thank you for your comment. We appreciate the reviewer’s insightful comments regarding the application and interpretation of Fisher’s exact test in Table 2. It is true that Fisher’s exact test is designed to assess whether there is a significant association among the categories of a variable, rather than pinpointing where specific differences lie. If any significant differences had been observed, further post hoc analyses would have been conducted to explore these differences in more detail.

  1. Provided point 3 above, one of my main concerns with this study is that authors undertook a longitudinal study. Thus, the appropriate tool to use to compare groups and timeframes is a between-within analysis of variance (or the less powerful strategy Bonferroni Correction). Could the authors argue about this?

Response: We acknowledge your concern regarding the statistical approach for comparing groups and time points. Our statistician re-conducted the analysis using linear mixed effect mode. We updated the results in table 5.

  1. My second main concern is that authors state that their findings suggest that “breast cancer survivors benefited from acupuncture…” [Line 303]. The statistical analysis performed assesses only differences, and not if one method is better than the other. I believe that authors missed to explicitly include their statistical hypotheses and the calculation of the corresponding effect sizes.

Response: Since we conducted mixed effect model and compare the symptom changes between two groups at week 6 and week 12, we can conclude that compared to usual care, acupuncture group had significant improvement in most of the symptoms.

This manuscript represents findings from an NIH-funded trial, and as per NIH/NCCIH guidance, the primary focus is on feasibility and acceptability rather than calculating the corresponding effect size (refer to PAR-21-240). We only explore the changes of symptom between two groups as secondary/exploratory aims.

Reviewer 2 Report

Comments and Suggestions for Authors

The title

Acupuncture is an interesting option in symptom management. This manuscript is about acupuncture for symptoms in medically underserved breast cancer survivors and it is presented as a pilot study on feasibility and acceptability. However, much of the content is in fact on the effects. The effects are interesting, but the title should be rephrased to include something about effects/ efficacy and that it is a pilot study.

Background

In its current form, the main aim of the study is to evaluate feasibility and acceptability, but the background is mainly about symptoms and efficacy. This could be better balanced. Moreover, for readers who work in countries with tax-financed health care, the issue of underserved populations could be better addressed, otherwise it is unclear why this particular group was chosen.

Aims
The aims should be better defined (with primary aims and secondary aims) and the presentation of Results should match the aims.

Materials and methods

I appreciate the Flow chart, but is should be improved. Readers need to know that 300 participants were approached of which 62 (21%) accepted participation. In an integrative acupuncture study this is important, as one could assume that persons interested in acupuncture were more likely to accept the invitation (almost 80% declined participation and unless a drop-out analysis is presented, it is a reasonable guess that the drop-outs were not random, which is a limitation that should be addressed).
Moreover, according to the Flow chart, 31 + 31 patients were allocated AND RECEIVED acupuncture/ usual care, but according to Table 4, 31+31 were allocated but 28 RECEIVED acupuncture and 30 received usual care.  
Follow-up: according to the Results, 4 patients missed acupuncture sessions: this should be addressed in the Flow chart.

Methodology

The authors chose an acupuncture group and a “usual care-group". This should be addressed as a limitation for two reasons: acupuncture is something very tangible and may therefore produce a strong placebo effect. In many studies, sham acupuncture is used in the control group to minimize bias. Another limitation is that the “usual care-group” did receive no special attention, while the acupuncture group received their treatments at an integrative care clinic.  Integrative clinics are known to focus on psychosocial support. Ten visits, with or without acupuncture could produce an effect, especially on symptoms such as anxiety or sleep, but also on pain (e.g., several studies show that loneliness is associated with higher pain intensity). Therefore, 10 visits with acupuncture, compared to 10 equally long visits without acupuncture would have been more ideal. These shortcomings should be addressed.

Instruments/ questionnaires

These are not adequately presented. Preferably, they should be added as supplements and the figures should be explained. E.g., PROMIS is presented as “items on a 5-point Likert scale”, but as seen in Table 3, those values are obviously transformed as the mean values seem to be over 50. Maybe it is a 0-100 scale?
Also the other instruments used should be described in more detail. Moreover, the references do not seem adequate. The study uses a 29-item measure in breast cancer but reference 19 is about PROMIS short form for fatigue and reference 20 is about prostate cancer. Moreover, in reference 20, more than 50% were over the age of 65, while the mean age in the current study is 55 years. In the prostate cancer study the authors show that PROMIS is age-dependent (differences in outcome between those over and below the age of 65). A validated instrument is only valid in similar contexts; therefore the author should address this, or find better references.

Results

As the main aim in the current form is feasibility and acceptability, these aspects should be better presented, including tables.

The effectiveness of the treatment is presented but only by statistical significances, not by any measures of clinical significance. If the PROMIS was transformed to a 0-100 scale (which I assume): what was your estimated MID (minimally important difference)? E.g., when using the EORTC QLQ-30 (values transformed to a 0-100 scale), MID is estimated to be 5-10 points, moderate effects are in the range of 10-20 points and considerable effects are above 20 points. This is e.g. elaborated by Osoba et al. in original studies from 1998 (JCO) and in a recent publication from 2023 (JCO).
According to Table 4, the effects, although statistically significant, were in the range of -1.1 to -9.6. Therefore, clinical significance should be critically discussed.

Table 4: the figures should be checked. E.g. for Depression weeks 12-0: the 95%CI is 7.0 (i.e., obviously wrong, it is not even an interval). You should also check Anxiety week 6 to week 0: the 95%CI:s are overlapping (-7.9- -2.0 and -3.1 to 1.7) and still the p-value is significant?

Long term follow-ups are difficult in clinical studies. In this study, there was a follow-up at 12 weeks. How did the authors check that the medical treatment of the studied symptoms did not change over time, as even a small increase e.g. in analgesics may have a great impact on pain assessments? This should be discussed as a limitation in the Discussion.

Discussion

Much of the Discussion is about symptoms and effects of the treatment, which is OK if the study is also on efficacy. Otherwise, more focus should be on feasibility and acceptability.

Minor comment: why are black and white written in versals (Black and White)?

Author Response

The title

Acupuncture is an interesting option in symptom management. This manuscript is about acupuncture for symptoms in medically underserved breast cancer survivors and it is presented as a pilot study on feasibility and acceptability. However, much of the content is in fact on the effects. The effects are interesting, but the title should be rephrased to include something about effects/ efficacy and that it is a pilot study.

Response: We have shifted our focus more toward feasibility and acceptability throughout the manuscript. Specifically, we expanded the introduction to include additional context about access disparities to acupuncture and emphasized implementation strategies in the discussion. This manuscript represents findings from an NIH-funded trial, and as per NIH/NCCIH guidance, the primary focus is on feasibility and acceptability. Therefore, we would like to retain the current title as it highlights the foundational nature of this study, which lays the groundwork for future research.

Background

In its current form, the main aim of the study is to evaluate feasibility and acceptability, but the background is mainly about symptoms and efficacy. This could be better balanced. Moreover, for readers who work in countries with tax-financed health care, the issue of underserved populations could be better addressed, otherwise it is unclear why this particular group was chosen.

Response: Thank you for your feedback. We revised the Introduction section (Pg 2, Lines 65-87) to better align with the study’s focus on acupuncture feasibility and acceptability. Additionally, we clarified that underserved U.S. cancer survivors face significant barriers to acupuncture access, such as financial constraints and limited resources.

Aims
The aims should be better defined (with primary aims and secondary aims) and the presentation of Results should match the aims.

Response: We have clarified the aims to ensure alignment between the study objectives and the presentation of results. The primary aim is to assess the feasibility and acceptability of integrating acupuncture into clinics serving medically underserved breast cancer survivors for the treatment of multiple symptoms. The secondary aim is to evaluate changes in symptom outcomes and compare these changes between the acupuncture group and the usual care group.

Materials and methods

I appreciate the Flow chart, but is should be improved. Readers need to know that 300 participants were approached of which 62 (21%) accepted participation. In an integrative acupuncture study this is important, as one could assume that persons interested in acupuncture were more likely to accept the invitation (almost 80% declined participation and unless a drop-out analysis is presented, it is a reasonable guess that the drop-outs were not random, which is a limitation that should be addressed).

Response: We acknowledge that only 21% of the approached participants agreed to enroll in the study, which raises the possibility of selection bias. Those who chose to participate may have had a predisposed interest in or belief in the efficacy of acupuncture, potentially limiting the generalizability of our findings to broader populations, including those less familiar with or less inclined toward acupuncture. Additionally, as no formal analysis of the characteristics of those who declined participation was conducted, it remains unclear whether the dropouts were random or systematically different from participants. We agree that future studies should include a dropout analysis to better understand these dynamics and assess their impact on study outcomes.

Moreover, according to the Flow chart, 31 + 31 patients were allocated AND RECEIVED acupuncture/ usual care, but according to Table 4, 31+31 were allocated but 28 RECEIVED acupuncture and 30 received usual care.  
Follow-up: according to the Results, 4 patients missed acupuncture sessions: this should be addressed in the Flow chart.

Response: Thank you for pointing this out. We acknowledge the discrepancy between the flowchart and Table 4 regarding the number of participants who received acupuncture and usual care. We updated the flowchart to reflect the accurate numbers, including the 4 participants who missed acupuncture sessions. Additionally, we revised the follow-up sections in the flowchart to clearly separate the Week 6 and Week 12 follow-ups and adjust the numbers accordingly (Pg 6, Figure 1).

Methodology

The authors chose an acupuncture group and a “usual care-group". This should be addressed as a limitation for two reasons: acupuncture is something very tangible and may therefore produce a strong placebo effect. In many studies, sham acupuncture is used in the control group to minimize bias. Another limitation is that the “usual care-group” did receive no special attention, while the acupuncture group received their treatments at an integrative care clinic.  Integrative clinics are known to focus on psychosocial support. Ten visits, with or without acupuncture could produce an effect, especially on symptoms such as anxiety or sleep, but also on pain (e.g., several studies show that loneliness is associated with higher pain intensity). Therefore, 10 visits with acupuncture, compared to 10 equally long visits without acupuncture would have been more ideal. These shortcomings should be addressed.

Response: Thank you for your feedback. This study focused on the feasibility and acceptability of implementing acupuncture in a clinical setting, not its efficacy. Therefore, we compared the acupuncture group to a usual care group to reflect real-world conditions. We acknowledge as a limitation that the usual care group did not receive equal attention, which could have influenced outcomes in the acupuncture group. We have addressed this limitation in the manuscript and recommend that future studies include a usual care group with equal attention to better control for these factors (Pg 11, Lines 402-407).

Instruments/ questionnaires

These are not adequately presented. Preferably, they should be added as supplements and the figures should be explained. E.g., PROMIS is presented as “items on a 5-point Likert scale”, but as seen in Table 3, those values are obviously transformed as the mean values seem to be over 50. Maybe it is a 0-100 scale?
Also the other instruments used should be described in more detail. Moreover, the references do not seem adequate. The study uses a 29-item measure in breast cancer but reference 19 is about PROMIS short form for fatigue and reference 20 is about prostate cancer. Moreover, in reference 20, more than 50% were over the age of 65, while the mean age in the current study is 55 years. In the prostate cancer study the authors show that PROMIS is age-dependent (differences in outcome between those over and below the age of 65). A validated instrument is only valid in similar contexts; therefore the author should address this, or find better references.

Response: We acknowledge the need for clearer descriptions of the instruments and their scoring. We have revised the manuscript to provide more detail on the PROMIS-29 V1.0 instrument, including an explanation of the T-score transformation (mean of 50, SD of 10) (Pg 5, Lines 186-189). Additionally, we updated the references to ensure they align with the use of PROMIS in similar context (age group) (Pg 5, Lines 186).

Results

As the main aim in the current form is feasibility and acceptability, these aspects should be better presented, including tables.

Response: We acknowledge the need for clearer presentation of our main aim. In response, we have added Table 3 summarizing acceptability outcomes to enhance clarity and alignment with the study's primary aim (Pg 8, Lines 282 and Table 3).

The effectiveness of the treatment is presented but only by statistical significances, not by any measures of clinical significance. If the PROMIS was transformed to a 0-100 scale (which I assume): what was your estimated MID (minimally important difference)? E.g., when using the EORTC QLQ-30 (values transformed to a 0-100 scale), MID is estimated to be 5-10 points, moderate effects are in the range of 10-20 points and considerable effects are above 20 points. This is e.g. elaborated by Osoba et al. in original studies from 1998 (JCO) and in a recent publication from 2023 (JCO).
According to Table 4, the effects, although statistically significant, were in the range of -1.1 to -9.6. Therefore, clinical significance should be critically discussed.

Response: We have added an explanation on the Minimally Important Change (MIC) in the Discussion section (Pg 11, Lines 363-367). However, unlike the EORTC measures, the MIC for PROMIS measures can vary widely depending on the specific domain and population studied. We reviewed various studies assessing MIC in cancer populations, and the MIC values reported were different across studies. A systematic review of MIC estimates for PROMIS measures suggests a range of 2 to 6 T-score points, and this range is considered to reflect clinically meaningful changes in patient-reported outcomes. We used 2-6 T score points to guide the discussion.

Table 4: the figures should be checked. E.g. for Depression weeks 12-0: the 95%CI is 7.0 (i.e., obviously wrong, it is not even an interval). You should also check Anxiety week 6 to week 0: the 95%CI:s are overlapping (-7.9- -2.0 and -3.1 to 1.7) and still the p-value is significant?

Response: Thank you for bringing this to our attention. We have carefully reviewed and corrected the figures in Table 4 (Pg 9, Table 5). Additionally, the p-value in this table indicates whether there are significant differences between the acupuncture group and the control group for each symptom measure.

Long term follow-ups are difficult in clinical studies. In this study, there was a follow-up at 12 weeks. How did the authors check that the medical treatment of the studied symptoms did not change over time, as even a small increase e.g. in analgesics may have a great impact on pain assessments? This should be discussed as a limitation in the Discussion.

Response: We acknowledge this concern and have added it as a limitation in the Discussion section (Pg 11, Lines 405-407).

Discussion

Much of the Discussion is about symptoms and effects of the treatment, which is OK if the study is also on efficacy. Otherwise, more focus should be on feasibility and acceptability.

Response: We have revised the Discussion section to place greater emphasis on the primary aims of the study, which are feasibility and acceptability. While we briefly address symptom outcomes as part of the secondary aim, the majority of the Discussion now focuses on the feasibility of integrating acupuncture into clinics serving medically underserved breast cancer survivors and the acceptability of this intervention among participants (Pg 10, Lines 334-345).

Minor comment: why are black and white written in versals (Black and White)?

Response: The AMA Manual of Style currently recommends capitalizing both Black and White when referring to racial and ethnic identities. Therefore, we are using capital letters.